# The spatial relation of diabetic retinal neurodegeneration with diabetic retinopathy

Jacoba A. van de Kreeke[1¤]*, Stanley Darma[2], Jill M. P. L. Chan Pin Yin[1], H. Stevie Tan[1,2], Michael D. Abramoff[3,4,5], Jos W. R. Twisk[6], Frank D. Verbraak[1,2]

1 Department of Ophthalmology, Amsterdam UMC, Location VUmc, Amsterdam, The Netherlands,
2 Department of Ophthalmology, Amsterdam UMC, Location AMC, Amsterdam, The Netherlands,
3 Department of Ophthalmology and Visual Sciences, University of Iowa Hospital & Clinics, Iowa City, Iowa, United States of America, 4 VA Medical Center, Iowa City, Iowa, United States of America, 5 IDx, Iowa City, Iowa, United States of America, 6 Department of Epidemiology and Biostatistics, Amsterdam UMC, location VUmc, Amsterdam, The Netherlands

¤ Current address: Department of Ophthalmology, Amsterdam UMC, Location VUmc, Amsterdam, The Netherlands
* ja.vandekreeke@amsterdamumc.nl

## Abstract

### Purpose

Diabetic retinal neurodegeneration (DRN) has been demonstrated in eyes of patients with diabetes mellitus (DM), even in the absence of diabetic retinopathy (DR). However, no studies have looked at the rate of change in retinal layers and presence/development of DR over time per quadrant of the macula. In this longitudinal study, we aimed to clarify whether the rate of DRN is associated with the development/presence of DR within 4 different quadrants of the retina.

### Methods

80 eyes of 40 patients with type 1 DM and no/minimal DR were included. At 4 visits over 6 years, SD-OCT and fundus images were acquired. Thickness of the Retinal Nerve Fiber Layer (RNFL), Ganglion Cell Layer (GCL) and Inner Plexiform Layer (IPL) was measured in a 1-6mm circle around the fovea overall and for each quadrant (superior, nasal, inferior, temporal). Fundus images were scored for the presence/absence of DR in these areas. Multi-level analyses were performed to determine the rate of change for each layer overall and per quadrant for eyes/quadrants without and with DR during the follow-up period.

### Results

RNFL and GCL showed significant thinning over time, IPL significant thickening. These changes were more pronounced for GCL and IPL in eyes/quadrants with DR during the follow-up period.

### Conclusions

RNFL and GCL both showed thinning over time, which was more pronounced in eyes with DR for GCL. This holds true even in regional parts of the retina, as quadrant analyses

**Data Availability Statement:** All relevant data are within the manuscript and its Supporting Information files.

**Funding:** MDA: NIH P30 EY025580, R01 EY018853, R01 EY019112 (https://nei.nih.gov/about/news-and-events/news), Research to Prevent Blindness (https://www.rpbusa.org/rpb/?) FDV: Supported by the Edward en Marianne Blaauwfonds, Netherlands Organization for Health Research and Development (https://www.zonmw.nl/en/), National Institutes of Health/National Eye Institute Grant R01-EY017066 (https://nei.nih.gov/about/news-and-events/news), and Research to Prevent Blindness (https://www.rpbusa.org/rpb/?). The funders had no role in study design, data collection and analysis, decision to publish, or preparation of the manuscript. IDx provided support in the form of salaries for author MDA, but did not have any additional role in the study design, data collection and analysis, decision to publish, or preparation of the manuscript. The specific roles of this author is articulated in the 'author contributions' section.

**Competing interests:** Co-author Michael D. Abramoff is employed by the company 'IDx'. This does not alter our adherence to PLOS ONE policies on sharing data and materials. IDx did not play a role in the study design, data collection and analysis, decision to publish, or preparation of the manuscript.

showed similar results, showing that structural DRN is associated with DR per quadrant independently.

## Introduction

Diabetic retinopathy (DR) is one of the leading causes of low vision and blindness in the Western World [1, 2]. Approximately one-third of the 285 million people with Diabetes Mellitus (DM) worldwide suffer from DR, and the prevalence of Diabetes Mellitus (DM) is expected to increase to about 552 million in 2030 [3]. This makes DR a serious health problem in ophthalmology, which is only expected to increase in the oncoming years [4]. Early detection and frequent monitoring of the retina is necessary to ensure adequate and timely treatment [4].

Since its development in 1991, optical coherence tomography (OCT) has become an indispensable diagnostic tool within the ophthalmological field [5, 6]. Based on the interference of reflected light, a detailed cross-sectional image of the retina can be obtained [5]. This enables visualization of pathological processes such as fluid or epiretinal membranes, but also permits measuring of the thickness of the retina and its layers [7].

Several studies have shown that retinal thickness is reduced in individuals suffering from DM, even in the absence of vascular DR, a process we have called Diabetic Retinal Neurodegeneration (DRN) [2, 8–12]. It is theorized this may be because chronic hyperglycaemia (or alternatively fluctuating levels of serum glucose), oxidative stress and accumulation of advanced glycation end products leads to an increase in glutamate and a loss of neuroprotective factors, which in turn causes neurodegeneration [2, 8, 9, 11, 12]. Especially the inner (neuronal) layers of the retina seem to be affected in DRN, in particular the retinal nerve fiber layer (RNFL), ganglion cell layer (GCL) and inner plexiform layer (IPL) [8, 9]. A classification scheme that combines all three retinal complications of diabetes—retinopathy proper, DRN, and diabetic macular edema—is under development, and a pilot study showed that DRN can be slowed down with intravitreal steroids [13, 14]. These studies showed that DRN in fact precedes any form of clinically detectable vasculopathy related to DR, and has the potential to serve as a very early detection method for retinal damage in DM, provided there is a causal or epiphenomenal relationship between DRN and vasculopathy. However, many of these studies are of a cross-sectional type, making relation statements about cause and effect difficult. It also has to be noted that the EUROCONDOR study found that, while there was indeed a clear association between DRN and vasculopathy, there is also a significant amount of diabetic patients not expressing this association [15].

Up till now, no studies have looked into the temporal and spatial relation between the onset of structural DRN and the onset of DR. However, such a relation has been shown for functional DRN, reporting a local change in mfERG with subsequent local DR development by Bearse et al [16]. Does the development of increased DRN imply DR is to be expected as a next or simultaneous step, or will this also occur in individuals who never develop DR? Are there regional differences in the relationship between DRN and DR? To gain more insight into these topics, we performed a longitudinal study in patients with type 1 DM and no or very early DR, to explore if the development of DRN is associated with the development or presence of DR, in 4 different quadrants (superior, nasal, inferior, and temporal) of the central retina.

## Methods

### Participants

This prospective cohort study consists of 40 patients with type 1 DM, also described previously [9, 10, 17–19]. Patient recruitment and follow-up took place from 2004 to 2014 at the

outpatient clinic of Internal Medicine at the Amsterdam UMC, location AMC. In 2007 the original OCT device (Zeiss Stratus, time domain OCT) was replaced by a spectral domain OCT (SD-OCT) device (Topcon). The present study is based on the SD-OCT measurements made during the last 7 years of follow-up only, and used 4 visits of each participant, covering a 5 year follow-up period for each individual participant. Some participants only underwent 3 visits over this period, causing the number of participants per visit moment to fluctuate slightly (all participants underwent the baseline visit). The study was approved by the Medical Ethics committee of the Amsterdam UMC, location AMC. The study followed the Tenets of the Declaration of Helsinki and written informed consent was obtained from all participants.

Inclusion criteria were: patients suffering from type 1 DM with no or mild non-proliferative DR (microaneurysms only, classified as no or mild DR according to the ICDR) as determined by a retinal specialist through indirect fundoscopy and fundus images [20].

Exclusion criteria were: more than mild DR (moderate, severe or very severe non-proliferative DR, proliferative DR, diabetic macular edema), refractive error >+5D or <-8D, best corrected visual acuity (BCVA) ≥0.1 LogMAR, severe media opacities interfering with scan quality, a history of ocular surgery other than cataract extraction, glaucoma, (a history of) uveitis or other retinal diseases interfering with retinal thickness (e.g. vitreoretinal interface pathology).

## Study visits

All participants underwent the following ophthalmological examinations at the study visits: best corrected visual acuity, intra-ocular pressure, refraction data, slit lamp examination, indirect fundoscopy, fundus photography and SD-OCT (Topcon 1000). Tropicamide 0.5% was used for pupil dilation to enable these examinations. Date of diagnosis of diabetes was obtained from charts, and time since diagnosis was calculated from this. HbA1C was calculated per visit from all HbA1C measurements in the preceding year. For final analyses, these HbA1C measurements were averaged to obtain 1 value for the whole follow-up period for each participant.

## Optical coherence tomography

SD-OCT (Topcon 3D OCT-1000, Topcon Medical Systems, Inc., Oakland, CA, USA) was used to image the participants. A 3-D volume scan protocol (6 × 6 × 2.3 mm), consisting of 128 B-scans, with 512 A-scans, each consisting of 650 measuring points, was used. All scans were segmented using the 3D segmentation algorithm developed by the Iowa Institute for Biomedical Imaging [21, 22]. Iowa Reference Algorithms is a free software program available in the public domain at https://www.iibi.uiowa.edu/content/shared-software-download. Individual layer thickness of RNFL, GCL and IPL in all 4 quadrants (superior, nasal, inferior, and temporal) in both the inner and the outer ring according to the Early Treatment Diabetic Retinopathy Study (ETDRS) macular grid were obtained for analyses. Values in quadrants of the inner and outer ring were averaged to obtain a single value for 4 regions from 1-6mmm in the macular area: superior, nasal, inferior an temporal (Fig 1). Because the surface area differs between the inner and outer ring (with a factor of around 3), this average calculation emphasized the changes in the inner ring, where, according to literature, DNR is most present [23].

## Fundus photography

Fundus images were obtained from both eyes with a 50˚ field of view using a TRC-50IX fundus camera; (Topcon Corporation, Tokyo, Japan). All images were graded for the presence or absence of DR by 2 trained graders (JAvdK and JMPLCPY). DR presence or absence was also graded in each quadrant (superior, nasal, inferior and temporal) around the fovea, as defined

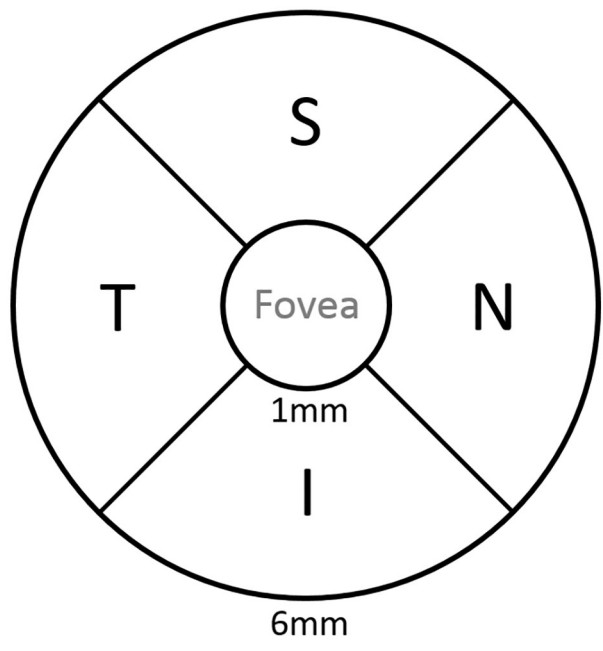

**Fig 1. Macular ETDRS grid used for defining the 4 quadrants for a right eye.** For a left eye, the T and N quadrants are vertically mirrored. S = Superior quadrant, N = Nasal quadrant, I = Inferior quadrant, T = Temporal quadrant.

in the OCT images (Fig 1). From the DR scores over 4 time points, a dichotomous variable was established consisting of either no DR (i.e. no DR on baseline or any of the subsequent visits) or development/presence of DR (i.e. no DR on baseline and DR on any of the subsequent visits, or DR on baseline and none or only some of subsequent visits). This was done for both the entire retinal region captured in the image and in each of the four quadrants individually within that image.

## Statistical analysis

To look at the differences in rate of change of retinal layer thickness over time in areas without DR versus those with development or presence of DR, we performed multilevel analyses consisting of 3 or 4 levels: quadrants were clustered within eyes (for the overall retinal layer thickness, this level was not included for the individual quadrant analyses), eyes were clustered within visits and visits were clustered within patients. Multilevel analyses enables correction for dependencies such as the expected dependencies between quadrants within eyes, eyes within the same visit and both eyes of an individual patient. We performed these analyses using both a crude and an adjusted model, corrected for age, sex, mean HbA1C and duration of DM (i.e. time since diagnosis). All analyses were performed using MLwiN (version 2.28, Center for Multilevel Modelling, University of Bristol, United Kingdom). Bar charts were drawn using Graphpad Prism (version 7.04, GraphPad Software Inc., California, USA).

## Results

Table 1 shows the descriptive information for the study population at each visit.

Tables 2 (crude) and 3 (adjusted) show the results of the multilevel analyses. In both models, RNFL and GCL showed a significant thinning over time, whereas IPL showed a slight

**Table 1. Demographics of the study population at each visit, data are means unless otherwise specified.**

|  | Visit 1 | Visit 2 | Visit 3 | Visit 4 |
|---|---|---|---|---|
| Number of participants (N) | 40 | 37 | 37 | 35 |
| Number of eyes (N) | 80 | 74 | 74 | 70 |
| Age (years) | 33.1 (±10.1) | 35.4 (±10.1) | 37.0 (±10.2) | 38.3 (±9.9) |
| Sex, N female (%) | 28 (70.0%) | 26 (70.3%) | 26 (70.3%) | 24 (68.6%) |
| Time since baseline visit (months) | - | 22.0 (±7.6) | 44.1 (±14.0) | 74.0 (±12.2) |
| DR N eyes (%) | 35 (43.8%) | 30 (39.0%) | 28 (36.4%) | 32 (42.7%) |
| BCVA (LogMAR) | -0.06 (±0.09) | -0.07 (±0.09) | -0.06 (±0.10) | -0.05 (±0.08) |
| HbA1C (mmol/mol, normal = <53) | 75.7 (±11.8) | 73.6 (±11.7) | 65.5. (±14.6) | 66.2 (±10.1) |
| Time since diagnosis of DM (years) | 18.7 (±8.5) | 20.7 (±8.5) | 22.4 (±8.4) | 23.6 (±7.6) |

Note that the individuals returning for a subsequent visit are not necessarily the same individuals returning the next visit (i.e. some skipped visit 2 only whereas others skipped visit 3 only etc.). DR = presence of Diabetic Retinopathy, BCVA = Best Corrected Visual Acuity (both eyes averaged), DM = Diabetes Mellitus.

thickening. For the GCL and IPL, more pronounced changes were associated with the presence or development of DR in those eyes/quadrants during the follow-up period, with overall and temporal GCL reaching a statistically significant difference. Fig 2 shows bar charts for the results obtained with the adjusted model.

## Discussion

This study, for the first time, shows that spatially, regional, *structural* DRN is associated with regional DR, confirming earlier studies that regional *functional* DRN is associated with regional DR. In addition, this study confirms earlier studies that DRN occurs irrespective of DR [8, 10, 11]. RNFL and GCL both showed a significant thinning over time, IPL showed a

**Table 2. Multilevel analysis for change in retinal layer thickness overall and per quadrant in years for both absence or presence of Diabetic Retinopathy.**

|  | No diabetic retinopathy | | | Diabetic retinopathy | | |
|---|---|---|---|---|---|---|
|  | Change in μm/year | p-value | SE | Change in μm/year | p-value | SE |
| RNFL overall | **-0.206** | **<0.001** | **0.023** | -0.179 | <0.001 | 0.035 |
| • Superior | -0.077 | 0.348 | 0.082 | -0.199 | 0.070 | 0.110 |
| • Nasal | **-0.292** | **<0.001** | **0.077** | **-0.386** | **0.002** | **0.124** |
| • Inferior | **-0.174** | **0.020** | **0.075** | -0.092 | 0.483 | 0.131 |
| • Temporal | -0.301[a] | <0.001 | 0.085 | -0.031[a] | 0.768 | 0.105 |
| GCL overall | -0.102[a] | 0.013 | 0.041 | -0.292[a] | <0.001 | 0.062 |
| • Superior | -0.127 | 0.140 | 0.086 | **-0.286** | **0.013** | **0.115** |
| • Nasal | -0.151 | 0.059 | 0.080 | **-0.301** | **0.021** | **0.130** |
| • Inferior | -0.121 | 0.121 | 0.078 | -0.220 | 0.108 | 0.137 |
| • Temporal | 0.011[a] | 0.901 | 0.088 | -0.331[a] | **0.003** | **0.110** |
| IPL overall | **0.084** | **0.016** | **0.035** | **0.166** | **0.001** | **0.052** |
| • Superior | 0.104 | 0.115 | 0.066 | 0.126 | 0.157 | 0.089 |
| • Nasal | 0.087 | 0.161 | 0.062 | 0.130 | 0.194 | 0.100 |
| • Inferior | 0.099 | 0.105 | 0.061 | 0.131 | 0.216 | 0.106 |
| • Temporal | 0.056 | 0.410 | 0.068 | **0.221** | **0.009** | **0.085** |

μm = micrometer, SE = Standard Error, RNFL = Retinal Nerve Fiber Layer, GCL = Ganglion Cell Layer, IPL = Inner Plexiform Layer.

[a] values in change/year differed significantly at p<0.05 between no DR and DR.

**Table 3. Multilevel analysis for change in retinal layer thickness overall and per quadrant in years for both absence or presence of Diabetic Retinopathy.**

| | No diabetic retinopathy | | | Diabetic retinopathy | | |
|---|---|---|---|---|---|---|
| | Change in μm/year | p-value | SE | Change in μm/year | p-value | SE |
| RNFL overall | *-0.206* | *<0.001* | *0.023* | -0.179 | <0.001 | 0.035 |
| • Superior | -0.079 | 0.336 | 0.082 | -0.203 | 0.065 | 0.110 |
| • Nasal | *-0.294* | *<0.001* | *0.077* | *-0.391* | *0.002* | *0.124* |
| • Inferior | *-0.175* | *0.020* | *0.075* | -0.098 | 0.454 | 0.131 |
| • Temporal | *-0.303*[a] | *<0.001* | *0.085* | -0.035[a] | 0.739 | 0.105 |
| GCL overall | *-0.105*[a] | *0.010* | *0.041* | *-0.294*[a] | *<0.001* | *0.062* |
| • Superior | -0.130 | 0.131 | 0.086 | *-0.288* | *0.012* | *0.115* |
| • Nasal | -0.154 | 0.054 | 0.080 | *-0.304* | *0.018* | *0.129* |
| • Inferior | -0.124 | 0.112 | 0.078 | -0.224 | 0.102 | 0.137 |
| • Temporal | 0.008[a] | 0.927 | 0.088 | *-0.332*[a] | *0.003* | *0.110* |
| IPL overall | *0.081* | *0.021* | *0.035* | *0.163* | *<0.001* | *0.052* |
| • Superior | 0.101 | 0.126 | 0.066 | 0.123 | 0.167 | 0.089 |
| • Nasal | 0.084 | 0.175 | 0.062 | 0.128 | 0.201 | 0.100 |
| • Inferior | 0.097 | 0.112 | 0.061 | 0.127 | 0.231 | 0.106 |
| • Temporal | 0.053 | 0.436 | 0.068 | *0.219* | *0.010* | *0.085* |

Corrected for age, sex, HbA1C and time since diagnosis of Diabetes Mellitus. μm = micrometer, SE = Standard Error, RNFL = Retinal Nerve Fiber Layer,

GCL = Ganglion Cell Layer, IPL = Inner Plexiform Layer.

[a] values in change/year differed significantly at p<0.05 between no DR and DR.

slight but significant thickening over time. For GCL and IPL, more pronounced changes were associated with the presence or development of DR in those eyes/quadrants.

Earlier studies showed that retinal (layer) thinning occurs naturally over time, likely due to an aging effect. Thinning of 0.01–0.16μm per year of the macular RNFL, 0.05–0.10μm of the GCL of and 0.05μm of the IPL have been described, and this thinning was more prominent in individuals of higher age [23–25]. In our DM population, we found an overall thinning of 0.206μm per year for the macular RNFL and 0.105μm for the GCL, and a thickening of 0.081μm per year of the IPL, after correction for multiple confounders in the eyes not suffering from DR. These values were 0.179μm, 0.294μm and 0.163μm respectively for the eyes with the presence/development of DR (Table 3). The thinning of the RNFL and GCL both exceed what was reported by other studies as occurring with physiological degeneration through aging, especially considering the younger age (mean age 33 years at baseline) of our study population. This confirms many earlier studies claiming that DM induces increased DRN (i.e. thinning) of the inner retinal layers, even in the absence of DR [8–11, 18, 26, 27]. Thickening of the IPL, however, is not known to be a results of aging, nor could we find this in studies looking at the effect of DM on the retina. In fact, the opposite has been described: IPL thinning was also reported in the eyes of individuals with DM [8, 10, 18, 26, 27]. However, most of these studies considered the GCL and IPL as one layer, due to the difficulty of correctly separating these two layers, especially with the early lower resolution time domain OCTs [10, 18, 26]. By analyzing this GCIPL complex as a whole, a slight thickening of the IPL in diabetics may be masked by a more pronounced thinning of the GCL. The slight, but significant thickening of the IPL in this longitudinal study may be due to very early leakage, causing microscopic fluid assembly in the inner plexiform layer, leading to increased thickness of this layer. Subclinical macular edema has mostly been described to occur in the inner nuclear layer, but neighboring layers are also thought to be affected, supporting this theory [26]. When eliminating the studies looking at the

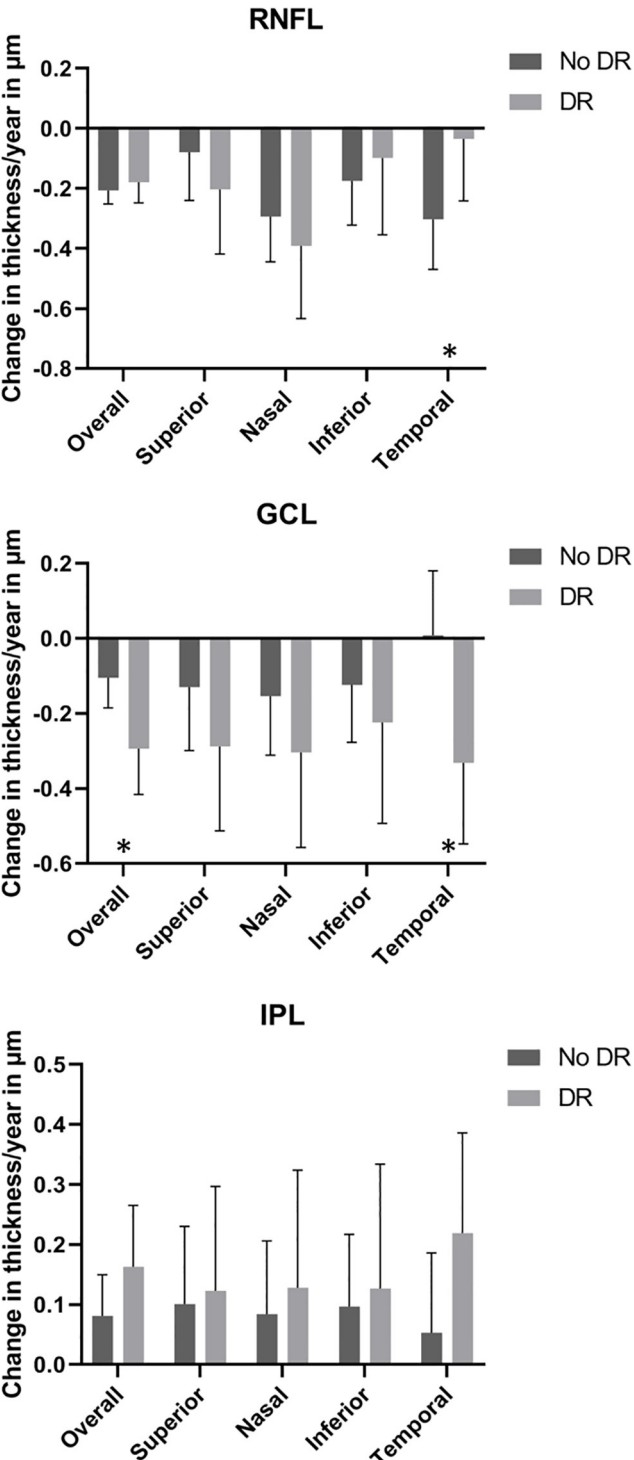

**Fig 2. Bar charts for change in retinal layer thickness overall and per quadrant in years for both absence or presence of Diabetic Retinopathy (DR).** Note that per quadrant means that that particular quadrant did not have any DR during the follow-up period, but other quadrants within that same eye may have had DR. Means and 95% confidence intervals obtained with multilevel analyses, corrected for age, sex, HbA1C and time since diagnosis of Diabetes Mellitus. RNFL = Retinal Nerve Fiber Layer, GCL = Ganglion Cell Layer, IPL = Inner Plexiform Layer. * values in change/year differed significantly at p<0.05 between no DR and DR.

GCIPL complex as a whole, only one study remains that found thinning of the IPL selectively, but only in a cross-sectional set-up when compared to healthy controls, and only in the eyes suffering from mild DR [27]. It could be that the IPL is thinner in DM patients with DR than controls, but IPL thickening over time may reflect a progress of leakage resulting in the build-up of subclinical edema. Furthermore, IPL thickening may also be caused by glial cell activation due to early inflammation, increasing the volume of these cells and therefore ultimately the volume of certain retinal layers. Lastly, the thickening of the IPL may be the result of an artifact from the segmentation algorithm, although this algorithm has been validated [21, 22].

In the overall values of both the GCL and the IPL, the eyes/quadrant with more pronounced changes were associated with the presence or development of DR during the follow-up period. For the GCL, this even reached a statistically significant difference. This suggests DR and DRN can develop in tandem. The finding that DR eyes show more severe DRN is one that is supported by many other studies [8, 9, 17, 18, 27]. However, the EUROCONDOR study did find that approximately 1 out of 3 diabetic patients (32%) do not express DRN despite the presence of minor DR, suggesting that while there seems to be a link in the pathogenesis of the two, this is not the same for all patients [15].

In this study, we also looked regionally in 4 different quadrants, to see if the finding of more pronounced DRN in areas with (development of) DR still holds true when analyzing these 4 quadrants within an eye. We found that the quadrants of the GCL showing increased DRN also suffered from or developed DR over the course of this study, with the temporal quadrant even reaching a statistically significant difference. This means that locally increased DRN is associated with an increased risk/presence of DR locally, i.e.: quadrants that suffer from increased DRN seem to suffer from DR more often, regardless of the overall DR status of that eye as a whole. This confirms an earlier functional DRN study using multifocal ERG, that the association between DR and neuroretinal degeneration is spatially limited and may fluctuate even within one eye [16]. In the case of the IPL, the quadrants with DR also had a higher increase in thickness over time, suggesting that the process of subclinical macular edema is also one that occurs very locally.

Both the overall RNFL changes and the RNFL changes per quadrant did not differ consistently between eyes/quadrants suffering from DR and those without, suggesting that RNFL is less influenced by the DR status in people with diabetes.

One of the main strengths of this study lies in its longitudinal set-up. This enables accurate estimations of retinal changes over time in diabetic patients, which are much less influenced by natural variation of retinal thickness or confounders such as age and sex. This is also illustrated by our use of both an uncorrected model and a model corrected for several confounders. The corrected model generally showed similar results as the uncorrected model, suggesting the confounders (age, sex, HbA1C and time since diagnosis of DM) to have little effect on the rate of change over time.

The lack of a control group in this longitudinal study is one of its limitations. Although several studies report normative data for retinal layer degeneration due to aging effects, a control group that underwent the exact same scanning protocol as our diabetic population would likely have given a more accurate insight in the added degeneration due to diabetes.

In conclusion, this study shows an association between spatially, regional, structural DRN and regional DR. We found a significant decrease of thickness over time of the RNFL and GCL, and an increase of thickness of the IPL in eyes of diabetic patients. The eyes/quadrants of the GCL and IPL with more pronounced DRN changes more often suffered from (development of) DR during the follow-up period, confirming DRN to develop even in eyes without DR, but once DR develops, this process is faster.

## Supporting information

**S1 Dataset. Dataset used for analyses of the data.**
(SAV)

## Author Contributions

**Conceptualization:** Jacoba A. van de Kreeke, Stanley Darma, H. Stevie Tan, Frank D. Verbraak.

**Data curation:** Stanley Darma, Jill M. P. L. Chan Pin Yin, Frank D. Verbraak.

**Formal analysis:** Jacoba A. van de Kreeke, Jos W. R. Twisk.

**Funding acquisition:** H. Stevie Tan, Frank D. Verbraak.

**Investigation:** Stanley Darma, Frank D. Verbraak.

**Methodology:** Jacoba A. van de Kreeke, Stanley Darma, Jill M. P. L. Chan Pin Yin, Michael D. Abramoff, Jos W. R. Twisk, Frank D. Verbraak.

**Software:** Michael D. Abramoff.

**Supervision:** H. Stevie Tan, Frank D. Verbraak.

**Writing – original draft:** Jacoba A. van de Kreeke.

**Writing – review & editing:** Stanley Darma, Jill M. P. L. Chan Pin Yin, H. Stevie Tan, Michael D. Abramoff, Jos W. R. Twisk, Frank D. Verbraak.

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
