## [Decision Letter · Decision Letter 0]

31 Dec 2019

PONE-D-19-29938

The spatial relation of diabetic retinal neurodegeneration with diabetic retinopathy

PLOS ONE

Dear Mrs van de Kreeke,

Thank you for submitting your manuscript to PLOS ONE. After careful consideration, we feel that it has merit but does not fully meet PLOS ONE’s publication criteria as it currently stands. Therefore, we invite you to submit a revised version of the manuscript that addresses the points raised during the review process.

Both reviewers found your paper interesting and convincing, but please address the concerns of the first reviewer concerning your multivalent modeling and please address the concerns of both reviewers about whether or not neurodegeneration can predict the appearance of microvascular abnormalities in the natural history of diabetic retinopathy.

We would appreciate receiving your revised manuscript by Feb 14 2020 11:59PM. To enhance the reproducibility of your results, we recommend that if applicable you deposit your laboratory protocols in protocols.io, where a protocol can be assigned its own identifier (DOI) such that it can be cited independently in the future. For instructions see: http://journals.plos.org/plosone/s/submission-guidelines#loc-laboratory-protocols

We look forward to receiving your revised manuscript.

Kind regards,

Alfred S Lewin, Ph.D.

Academic Editor

PLOS ONE

Journal Requirements:

2. Please disclose the author affiliation to IDx in your competing interest statement.

i)  Thank you for stating the following in the Competing Interests section:

[The authors have declared that no competing interests exist.].   

We note that one or more of the authors are employed by a commercial company: "IDx".

Reviewers' comments:

Reviewer's Responses to Questions

**Comments to the Author**

1. Is the manuscript technically sound, and do the data support the conclusions?

Reviewer #1: Yes

Reviewer #2: Yes

2. Has the statistical analysis been performed appropriately and rigorously? 

Reviewer #1: Yes

Reviewer #2: Yes

3. Have the authors made all data underlying the findings in their manuscript fully available?

Reviewer #1: No

Reviewer #2: Yes

4. Is the manuscript presented in an intelligible fashion and written in standard English?

Reviewer #1: Yes

Reviewer #2: Yes

5. Review Comments to the Author

Reviewer #1: The authors present an interesting analysis of structural changes in time series of OCT data from diabetic patients. The aim of their work is clearly expressed. In general, the statistical analysis and the methodology is correct.

I have some concerns regarding the multilevel modelling:

- From their description at lines 151-153 it would look like they have employed 4 nested factors (quadrants, eye, visits and patient). However, quadrants and visits are effectively fixed effects in the analysis (calculations per quadrants are clearly reported in the table and the visit is effectively the same as the time variable in the analysis). The only two random effects should therefore only be patient and eye (nested)

- Were random slopes included in the analysis? Since each eye represents a different time series, having random slopes specific for each eye would greatly improve the accuracy of the estimates

One of the stated aims (and claimed results in the discussion) is to understand whether structural changes are associated with the (local) development of DR. The authors seem to imply in their discussion that they have gathered evidence for thinning of the inner retina as a prognostic factor for the development of DR. This is evident when they draw the parallelism with similar studies investigating the prognostic ability of functional measurements. However, their analysis does not focus on an accurate assessment of the risk of developing DR given the structural parameters. Indeed, their results, as presented, would be more correctly interpreted as faster loss of inner retinal neurons as a consequence of DR. The authors are unclear about what interpretation they want to support and this should be clarified.

Also, there is no mention of how many subjects (sectors?) actually converted to DR during the study. If the aim is to show that neural loss is associated with an increased risk of developing DR, or simply micro-vascular changes in a given sector, an analysis with a logistic regression (or better yet, a survival analysis able to incorporate the temporal dimension) should be performed using the structural parameters as a predictor and the conversion to DR as the response variable.

Reviewer #2: In this paper the relationship between diabetes-induced retinal neurodegeneration and microvascular changes in fundus photography is examined in a prospective observational study. The main conclusion is that RNFL and GCL showed significant thinning over time and these changes were more pronounced with the presence or development of DR. Although a control group is lacking, this is a very interesting study which add new clinical data regarding the relationship between retinal neurodegeneration and the development microvascular abnormalities in the setting of diabetic retinopathy. There are, however, several issues that should be addressed:

In the introduction and/or in the discussion, the results of EUROCONDOR study in which two phenotypes of early stages of DR: one with diabetes induced retinal neurodegeneration + microvascular abnormalities, and another in which microvascular abnormalities were detected without any sign of neurodegeneration should be mentioned. In this regard, the authors could comment on the percentage of patients who developed microvascular abnormalities without significant thinning in RNFL and/or GCL in SD-OCT.

The progression of microvascular abnormalities during follow-up and their relationship with neurodegeneration should be better analyzed. A separate analysis between new incident microvascular lesions and those already present at the beginning of the study could clarify the role of neurodegeneration on new onset of vascular abnormalities (at least in a subset of patients). The question of whether or not neurodegeneration can predict the appearance of microvascular abnormalities in the natural history of DR should be addressed by the authors taken into account the prospective nature of this study.

The second sentence of the first paragraph of the discussion (lines 203-204) needs to be completed with appropriated references.

Apart from vascular leakage, glial activation could also contribute to the thinning of IPL. The authors perhaps could add a comment on this issue to the discussion section.

6. PLOS authors have the option to publish the peer review history of their article (what does this mean?). If published, this will include your full peer review and any attached files.

Reviewer #1: No

Reviewer #2: No

---

## [Author Response · Author response to Decision Letter 0]

15 Jan 2020

There is an uploaded file regarding the response to reviewers.

---

## [Decision Letter · Decision Letter 1]

3 Feb 2020

PONE-D-19-29938R1

The spatial relation of diabetic retinal neurodegeneration with diabetic retinopathy

PLOS ONE

Dear Mrs van de Kreeke,

Thank you for submitting your manuscript to PLOS ONE. After careful consideration, we feel that it has merit but does not fully meet PLOS ONE’s publication criteria as it currently stands. Therefore, we invite you to submit a revised version of the manuscript that addresses the points raised during the review process.

The first reviewer indicates that that the sectors should not be treated as a random effect, and it is difficult to see how different sectors of the same eye can be treated as independent events.

We would appreciate receiving your revised manuscript by Mar 19 2020 11:59PM. To enhance the reproducibility of your results, we recommend that if applicable you deposit your laboratory protocols in protocols.io, where a protocol can be assigned its own identifier (DOI) such that it can be cited independently in the future. For instructions see: http://journals.plos.org/plosone/s/submission-guidelines#loc-laboratory-protocols

We look forward to receiving your revised manuscript.

Kind regards,

Alfred S Lewin, Ph.D.

Academic Editor

PLOS ONE

Reviewers' comments:

Reviewer's Responses to Questions

**Comments to the Author**

1. If the authors have adequately addressed your comments raised in a previous round of review and you feel that this manuscript is now acceptable for publication, you may indicate that here to bypass the “Comments to the Author” section, enter your conflict of interest statement in the “Confidential to Editor” section, and submit your "Accept" recommendation.

Reviewer #1: (No Response)

Reviewer #2: All comments have been addressed

2. Is the manuscript technically sound, and do the data support the conclusions?

Reviewer #1: Yes

Reviewer #2: (No Response)

3. Has the statistical analysis been performed appropriately and rigorously? 

Reviewer #1: Yes

Reviewer #2: (No Response)

4. Have the authors made all data underlying the findings in their manuscript fully available?

Reviewer #1: No

Reviewer #2: (No Response)

5. Is the manuscript presented in an intelligible fashion and written in standard English?

Reviewer #1: Yes

Reviewer #2: (No Response)

6. Review Comments to the Author

Reviewer #1: I appreciate that the authors provided explanation to some of my concerns. To my eyes, however, some sentences still read as confusing:

- Lines 245 - 246: "...thus making increased neurodegeneration a sign of potential worsening of retinal vascular status and vice versa in DM" should be removed

- Lines 255 - 258: "This means that locally increased DRN is associated with also constitutes an increased risk/presence for (development of) DR locally, i.e.: quadrants that suffer from increased DRN tend seem to develop or suffer from DR more often, regardless of the overall DR status of that eye as a whole." should be reworded to avoid suggesting a causative effect. The word "development" is especially misleading.

Finally, I am still convinced that the sector should not be a random effect. The author explicitly use the different sectors to make inference from direct comparisons, reporting p-values. This is exactly when a factor should be a fixed rather than a random effect. Moreover, having the same factor twice (as random and fixed effect) generates collinearity and is simply wrong. Numerical improvement of fitting should not replace good reasoning when building statistical models.

Reviewer #2: (No Response)

7. PLOS authors have the option to publish the peer review history of their article (what does this mean?). If published, this will include your full peer review and any attached files.

Reviewer #1: No

Reviewer #2: No

---

## [Decision Letter · Decision Letter 2]

11 Mar 2020

PONE-D-19-29938R2

The spatial relation of diabetic retinal neurodegeneration with diabetic retinopathy

PLOS ONE

Dear Mrs van de Kreeke,

Thank you for submitting your manuscript to PLOS ONE. After careful consideration, we feel that it has merit but does not fully meet PLOS ONE’s publication criteria as it currently stands. Therefore, we invite you to submit a revised version of the manuscript that addresses the points raised during the review process.

The reviewer still does not believe that you have accounted for differences between quadrants in the same subject. Please see below.

We would appreciate receiving your revised manuscript by Apr 25 2020 11:59PM. To enhance the reproducibility of your results, we recommend that if applicable you deposit your laboratory protocols in protocols.io, where a protocol can be assigned its own identifier (DOI) such that it can be cited independently in the future. For instructions see: http://journals.plos.org/plosone/s/submission-guidelines#loc-laboratory-protocols

We look forward to receiving your revised manuscript.

Kind regards,

Alfred S Lewin, Ph.D.

Academic Editor

PLOS ONE

Reviewers' comments:

Reviewer's Responses to Questions

**Comments to the Author**

1. If the authors have adequately addressed your comments raised in a previous round of review and you feel that this manuscript is now acceptable for publication, you may indicate that here to bypass the “Comments to the Author” section, enter your conflict of interest statement in the “Confidential to Editor” section, and submit your "Accept" recommendation.

Reviewer #1: (No Response)

2. Is the manuscript technically sound, and do the data support the conclusions?

Reviewer #1: Partly

3. Has the statistical analysis been performed appropriately and rigorously? 

Reviewer #1: Yes

4. Have the authors made all data underlying the findings in their manuscript fully available?

Reviewer #1: No

5. Is the manuscript presented in an intelligible fashion and written in standard English?

Reviewer #1: Yes

6. Review Comments to the Author

Reviewer #1: The authors have addressed all my concerns except the ones regarding the random effects in the mixed model. I appreciate the explanation, but what they have done does not account for differences between quadrants in the same subject/eye. If they wanted to model this, they should have used a random SLOPE for the effect of quadrants. This would model the INDIVIDUAL effect of quadrants within each eye/individual.

Allow me to elaborate with a simpler example (Quadrant is a discrete factor in the model):

y = B0 + B1*Quadrant + (1|Subject/Eye) #This models the quadrant as a fixed effect

y = B0 + B1*Quadrant + (1|Subject/Eye/Quadrant) #This models the quadrant as a fixed effect and random intercept,

nested within the eye. This model shows a bad use of random effects,

since the grouping factor is considered both as a random and a fixed

effect.

y = B0 + B1*Quadrant + (1|Quadrant/Subject/Eye) #This is useless

y = B0 + B1*Quadrant + (1|Subject/Eye) + (1|Quadrant) #This is equally useless

y = B0 + B1*Quadrant + (Quadrant|Subject/Eye) #THIS models individual effects of the quadrant within each eye/subject,

which is what the authors state as their goal for including the quadrant in

the random effects. In this case, Quadrant is a random slope on the fixed

effect.

7. PLOS authors have the option to publish the peer review history of their article (what does this mean?). If published, this will include your full peer review and any attached files.

Reviewer #1: No

---

## [Author Response · Author response to Decision Letter 2]

23 Mar 2020

Due to the concerns raised by the reviewer, we decided to remove the quadrant level. Tables 1 and 2, as well as figure 2, have been updated with the results obtained with the new analyses (i.e. analyses without the quadrant level). The methods section has also been adjusted accordingly (page 7, lines 153-154). As the results did not change drastically, the message our paper conveys did not change either, and therefore the discussion and conclusions required no alteration.

---

## [Editor Report · Decision Letter 3]

26 Mar 2020

The spatial relation of diabetic retinal neurodegeneration with diabetic retinopathy

PONE-D-19-29938R3

Dear Dr. van de Kreeke,

We are pleased to inform you that your manuscript has been judged scientifically suitable for publication and will be formally accepted for publication once it complies with all outstanding technical requirements.

With kind regards,

Alfred S Lewin, Ph.D.

Section Editor

PLOS ONE
---

## [Editor Report · Acceptance letter]

2 Apr 2020

PONE-D-19-29938R3 

The spatial relation of diabetic retinal neurodegeneration with diabetic retinopathy 

Dear Dr. van de Kreeke:

I am pleased to inform you that your manuscript has been deemed suitable for publication in PLOS ONE. Congratulations! Your manuscript is now with our production department. 

With kind regards,

on behalf of

Dr. Alfred S Lewin 

Section Editor

PLOS ONE